# An Integrated Review of Transitional Care for Families of Pre-Term Infants

**DOI:** 10.3390/healthcare12222287

**Published:** 2024-11-15

**Authors:** Jeong Soon Kim, Hae Ran Kim

**Affiliations:** 1Department of Nursing, Chungwoon University, Hongseong 32244, Republic of Korea; jskim2022@chungwoon.ac.kr; 2Department of Nursing, College of Medicine, Chosun University, Gwangju 61452, Republic of Korea

**Keywords:** pre-term infants, parents, transitional care, at home

## Abstract

This study was conducted to identify the key elements of transitional care for families with pre-term infants in South Korea. We used an integrative review methodology proposed by Whittemore and Knafle. During the problem identification step, the review question was constructed via the population, intervention, outcome, and timeframe (PIOT) format. During the literature search step, integrative reviews of the published literature from nine electronic databases were undertaken and a total of 14 studies were reviewed that met our inclusion criteria. During the data evaluation step, the quality of the literature was assessed using the Mixed Methods Appraisal Tool (MMAT) developed by Hong et al. We identified three domains and 10 key attributes of transitional care for families of pre-term infants. The parenting empowerment domain included growth and development, developmental promotion, nutrition, safe environment, general parenting, and parent–infant interaction. The emotional support domain included counseling, advocacy, and community network. The social support domain included professional collaboration. Efforts should continue to further develop programs and policies to enhance transitional care for families of pre-term infants that reflect South Korean nursing practices.

## 1. Introduction

Pre-term infants are defined as live-birth babies that are born before 37 weeks of gestation. In 2020, 13.4 million babies, or 10% of newborn births, were born prematurely [1], and Korea’s premature birth rate in 2022 was 9.8%—an increase of 1.6 times in 10 years [2]. Between 28 and 40 weeks of gestation, the fetus prepares for life after birth by accumulating subcutaneous fat, as well as further developing its central nervous, respiratory, and gastrointestinal systems [3]. Pre-term infants are often born with life-threatening health conditions and various growth and developmental conditions because they do not have enough time to properly develop in utero. Fortunately, owing to rapid medical advances and improved neonatal intensive care unit (NICU) care, survival rates have increased and the prognoses of pre-term infants have improved significantly in terms of severe health-related complications.

Although pre-term infants who are discharged from hospital have shown many improvements in health, they are often much weaker than full-term infants and require constant attention to address their various medical complications, as well as growth and developmental issues. Due to this, parents are often pleased to return home with their babies but may also be confused by the burden of caring for a fragile infant, feeling unprepared and experiencing vague fears [4]. Parents caring for pre-term infants at home may therefore seek out more specialized support throughout the process of raising a pre-term infant. Specifically, they often want to learn about proper feeding practices, any available expert support from the community or local healthcare providers, and creating a safe environment in which to raise a vulnerable child [5,6].

Transitional care focuses on helping pre-term infants prepare for the transition from the NICU to home, as well as initiating supportive developmental care at home. It includes general infant care, as well as specialized care for pre-term infants. The main goals of transitional care are to improve parenting and nurturing skills and to build parent–child attachments [7]. The main challenges faced in this practice are pre-discharge planning, education, home equipment, and home visits by healthcare providers. The latter are typically conducted through home visits that involve skilled healthcare personnel, including neonatal nurse practitioners [8,9].

In South Korea, research on nursing care for pre-term infants has been considered important and has been conducted continuously. Most related research topics have focused on NICU inpatient care and discharge education; therefore, there is a lack of research on transitional care [10]. In addition, there are not enough systems and policies for transitional care and few in-home support systems for families of pre-term infants [11,12]. As a result, parents of pre-term infants spend a great deal of personal energy managing their own care environment and acquiring the parenting skills necessary to raise a pre-term infant, which increases the stress and burden of parenting. Therefore, various studies and efforts are warranted to identify and establish transitional care for families with pre-term infants.

We aimed to identify the important elements of transitional care for families with pre-term infants in South Korea, to collect basic data for the development of related programs, and to suggest beneficial future directions of transitional care for families with pre-term infants suitable for domestic implementation.

## 2. Theoretical Framework

This study used the neonatal transitional care (NTC) model as a conceptual framework. This model is based on the philosophies of family-centered care (FCC) and transitional theory [13].

FCC is a philosophy and method of healthcare delivery that recognizes parents as experts in terms of their children’s needs, promotes partnerships between parents and healthcare providers, and supports the family’s role in decision making for their children [14]. Transition theory recognizes that transition is a psychological process of adapting to change that is most often associated with developmental stages, changes in health, and changes in the social environment [15]. Therapeutic interventions by nurses also play important roles in transitional theory [13,15].

NTC is a program model that supports parents in caring for their children following discharge from the NICU, because most premature infants are discharged from the NICU with health conditions that are more difficult to care for than those of full-term infants [13]. The main goals of NTC are whole-family involvement, facility and financial support, emotional support for the family, linkage to community health services, and information sharing [13]. NTC begins with several meetings prior to discharge, aimed at identifying the family’s needs and building a therapeutic relationship. This process continues after discharge through home visits or phone consultations. Highlights include infant assessment, development, feeding routines, medication administration, family health and wellness, and discussion of maternal depression [16]. The development of a specialized multidisciplinary team that includes midwives, neonatal nurses, medical staff, and ancillary personnel is essential for successful NTC implementation—as is the development of family contact protocols, including family visits and telephone counseling [13].

The benefits of NTC include improved maternal mental health, increased breastfeeding rates, enhanced confidence and ability to bond with the baby in parents, minimized length of NICU stay and reduced readmission rates, facilitated effective discharge planning for infants with complex conditions, and improved multidisciplinary collaboration [17].

In the Republic of Korea, FCC has been emphasized in clinical practice, including NICUs, and has been utilized as a basic theory for nursing interventions to improve mother–infant attachment during hospitalization and to emphasize the role of parents in discharge education [10,12]. However, transitional care at home after discharge is still inadequate [11]. Therefore, it is necessary to identify precedents of well-established home visiting nursing systems and parenting care programs using internet-based smart devices in other countries and consider them in detail [18,19]. In addition, institutional support for families of premature infants returning home after NICU discharge is needed to adapt the results to Korean conditions. This will be preceded by efforts to gradually develop programs based on transitional care, along with education to improve clinical practice understanding of transitional care.

## 3. Materials and Methods

### 3.1. Research Design

This study was designed following the integrative review format proposed by Whittmore and Knafl in 2005 [20]. It followed the format’s five stages to thoroughly explore developmental supportive care within the homes of families with pre-term infants, toddlers, and newborns. These five stages included: problem identification, literature search, data evaluation, data analysis, and presentation.

### 3.2. Research Procedure

#### 3.2.1. Step 1: Problem Identification

It is important to clearly identify the purpose of any literature review or study [20]. Such central questions can be formulated using the population, intervention, outcome, and timeframe (PIOT) format [21]. For this study, the population (P) refers to families with a pre-term infant in their home. The interventions (I) comprised the essential components of a transitional program needed for families with pre-term infants living at home. The outcome (O) measured health and well-being indicators in both parents and pre-term infants. The timeframe (T) was the time spent at home following discharge from the NICU.

The review question adopted was “What are the key components of transitional care for families with pre-term infants after NICU discharge?” (Table 1).

#### 3.2.2. Step 2: Literature Search

A literature search is the process of identifying all appropriate sources for a topic. It should include precise search terms, the databases used, additional search strategies, and inclusion and exclusion criteria to increase the rigor of the review [20].

The literature search period for this study was from June to November of 2023, and the eligible publication period was between 2012 and 2023. The databases searched included KISS, RISS, NDSL, and DBpia in South Korea, as well as Google Scholar, MEDLINE, Embase, CINAHL, and Science Direct abroad. We searched using the following search terms: pre-term infant, parents, developmental(ly), support care, in (at)-home, transitional care, home based, and discharge NICU. The phrase approach utilized a strategy of combining search terms and keywords with “AND”, “OR”, and the wildcard symbol (*) to include more relevant results.

##### * Inclusion Criteria

In this review, we included (1) studies of qualitative, quantitative, and mixed methods spanning articles, interventions, reports, components or strategies, reviews, and theses. (2) The included studies were those concerning infants and their parents at home after discharge from the NICU. (3) Papers written in both English and Korean were included, to address more of the available information.

##### * Exclusion Criteria

To ensure that our search was accurate, we excluded studies based on the following criteria: (1) those that included interventions for patients hospitalized in the NICU; (2) studies for which the full text was inaccessible; (3) studies such as other reviews, case studies, and case reports.

A total of 620 records were identified in our primary search. During the electronic database search, manual searches were performed in parallel to search for additional documents or studies that could be included. After excluding 491 papers that did not meet our inclusion criteria, 90 paper duplicates, 21 papers with unclear content (not enough transitional care content), and 4 papers that were deemed low-quality, a total of 14 papers were selected for inclusion (Figure 1).

#### 3.2.3. Step 3: Data Evaluation

The data evaluation step of an integrative review is the process of assessing the quality of the selected literature. Because there are no set standards for assessing and interpreting quality, methods for assessing quality vary depending on the study design [20].

In this study, the quality of the literature was assessed using the Mixed Methods Appraisal Tool (MMAT) developed by Hong et al. in 2019 [22]. The MMAT is an assessment tool designed to evaluate the quality of various research designs in the literature, including qualitative, quantitative, and mixed-method studies. To assess methodological quality, the MMAT (ver. 2018) consisted of two screening questions and 20 items: qualitative (5 questions), randomized controlled trial (5 questions), non-randomized (5 questions), quantitative descriptive (5 questions), and mixed methods (5 questions). Each question could be answered with “yes”, “no”, or “can’t tell” (in terms of reporting of unclear information related to the criterion), with “no” and “can’t tell” being considered low-quality papers.

During the quality appraisal process, two of the authors rigorously reviewed and analyzed the content of the 20 selected articles using MMAT. We conducted several online and offline meetings to discuss and reach a consensus. In any cases of disagreement, a doctor of nursing with experience in integrative reviews was consulted to make a final assessment. As a result, we eliminated 6 articles that did not meet our criteria for topic clarity and methodological appropriateness, including one article whose scope was limited to rehabilitation (1), three studies whose population was limited to children with health conditions (3), and two studies whose scope was limited to the time of NICU admission (2) [23,24,25,26,27,28]. The remaining 14 articles had clear research topics, research methods, and procedures; met the study objectives; and were free of statistical errors (Table 2).

#### 3.2.4. Step 4: Data Analysis and Presentation

The data analysis step is important for the innovative synthesis of data and unbiased interpretation of the primary literature. It should present tables or figures to demonstrate the logic of the data analysis [20]. The final 14 selected articles were reviewed iteratively to derive systematic categories through reduction, redefinition, comparison, and synthesis, which were then tabulated. We then organized the key attributes of transitional care for families with pre-term infants and presented them in another table.

## 4. Results

### 4.1. Characteristics of the 14 Included Articles

The publication years of the articles on transitional care for pre-term infants selected for this study were: 2012–2015 (5; 35.7%), 2016–2019 (6; 42.9%), and 2020–2022 (3; 21.4%). In terms of study design, a total of 11 (78.6%) quantitative studies were published, of which 7 (50.2%) were RCTs and 2 (2.2%) were quasi-experimental studies or cohort studies. Three (21.4%) qualitative studies explored parents’ feelings regarding raising their children at home after discharge from the NICU. Two of the qualitative studies [39,40] explored satisfaction and parental experiences with transitional care programs. One of the qualitative studies [38] focused on the experiences and care needs of parents caring for their pre-term infants at home after discharge from the NICU and did not include a transitional care program. Two of the studies [33,39] applied theoretical concepts, including ET and FCC (Table 3).

### 4.2. Methods of Transitional Care

The subjects of the studies included mothers, parents (both mothers and fathers), infants, and families (mother–infant). Of these, five [19,29,32,33,40] involved only mothers. The remaining nine included parents and families. We found that 11 (78.6%) of the reports focused on mothers, indicating that mothers represent a crucial component of the transition period. However, it is also noteworthy that the focus of parenting is not limited to mothers only, as we also saw an increasing number of parenting articles since 2020 that included fathers (Table 4).

The methods used for transitional care included home visits, telephone consultations, video consultations, and smartphone-based approaches. Four of the included studies used only home visits (two published in 2015 [35,36] and two published in 2017 [30,32]). Four studies combined home visits and telephone counseling (one published in each of 2012 [33], 2015 [29], 2016 [31], and 2022 [39]). One study, published in 2021, used video consultation [38], but had a qualitative design that explored experiences with transition interventions. Two studies implemented internet-based education and interventions (one published in 2013 [34] and one in 2016 [37]). Two studies used smartphone app-based interventions—conducted in 2016 [18] and 2022 [19]. Both were RCTs designed to prove the effectiveness of a smartphone-based transitional care program. Since 2012, the main medium of transitional care provision has been home visits and telephone consultations. It could thus be seen that the changes in the medium of care delivery paralleled the development of ICT technology.

The duration of transitional care ranged from 3 weeks to 12 months. The number of articles with a 1-month period was the highest (four articles [18,29,33,38]), followed by 3 months (three articles [32,34,35]), 6 months (two articles [36,39]), and 12 months (three articles [19,30,31]). The shortest intervention period was 3 weeks [37], using web applications and video interviews (Table 4).

### 4.3. Measurement of Transitional Care

The outcome variables used to test the effectiveness of each transitional care program varied significantly. In this study, the variables that showed significant results on effectiveness were summarized, and the variables were measured separately for parents (specifically mothers) and infants. The variables measured for parents included maternal anxiety [29,35], confidence in one’s parenting role [29], parenting knowledge [38], parenting satisfaction [34,37], parents’ quality of life [35], and parenting competence [18]. Most of these were measured using psycho-emotional indicators. For infants, whose levels of emotional development are incomplete, physical growth and developmental indicators were used. These indicators included physical health [35], development [31,36], mortality rate and immunization [30], rate of outpatient visits [30,32,37], and rate of illness [33] (Table 4).

### 4.4. Contents of Transitional Care

We identified three domains and 10 key attributes of transitional care programs for families with pre-term infants. The three domains were parenting empowerment, emotional support, and social support (Table 5). The goal of successful transitional care for families of pre-term infants is to ensure successful growth and development at home. Parenting skills that reflect the characteristics of their own child are a critical success factor in transitional care and can be achieved through systematic parent education that begins before discharge from the hospital. Emotional support from medical staffs through visiting and telephone counseling, as well as social support from public health centers and community healthcare professionals, played a nourishing role in strengthening parenting empowerments. We visualized the results, as shown Figure 2, to help the reader understand the key areas and interactions of the transition.

#### 4.4.1. Parental Empowerment

The largest component of the transitional care program was the educational intervention. These interventions aimed to improve the parenting skills of parents experiencing transitional care at home following discharge from the hospital. Of the 14 studies analyzed, 13 included educational interventions. All of these included information concerning child growth, development, and monitoring during transitional care. Two of the articles [18,37] included information related to understanding the characteristics and activities of pre-term infants. Skin-to-skin care [37], touching [29,30,31], sensory stimulation [35], and rehabilitation [39] were analyzed as interventions used to promote growth and development.

Nutrition was also discussed in most of the literature. Regarding nutrition, the most common topics were breastfeeding maintenance and management [29,32,34,36,37], as well as preparing nutrition [19,39]. Pre-term infants are less likely to be able to properly breastfeed in the beginning, owing to their NICU stays and physical characteristics. However, breast milk is the best source of nutrition for infants, and nursing practitioners strongly encourage parents to breastfeed. Therefore, it was expected that breastfeeding would be included in transitional care programs.

These programs also included a section concerning safe environments for pre-term infants. Safe environments are one of the primary concerns of most parents, as it is necessary for vulnerable infants to be raised in normal family environments rather than clinical settings. The difference between maintaining a safe environment in a typical infant care home is the inclusion of specialized medical equipment. Specifically, injury prevention [30], emergency care [33], medication, and nutritional support [30,36] may all make use of medical devices. Infant monitoring, O_2_ supply [29], follow-up scheduling [30], and environmental care [19,31,33] were also included in many transitional care programs. General infant care included bathing [19,29,36], safe sleep [19,39], addressing crying [30], and parent–infant interactions [31,35].

#### 4.4.2. Emotional Support

Transitional care can also include emotional support for parents. During discharge from the hospital and transition to home, the object of care remains the same, but its focus shifts from the healthcare professional to the parent. During this time, parents can experience emotional difficulties as they may feel unprepared to meet their child’s diverse needs. Therefore, support for parents in transitional care is essential. Six of our included articles (44.4%) included emotional support for parents. Transition program content in the experimental design literature included counseling and support [18,36], coping with care [34,36], comfort support and understanding care plans [32], and communication and sharing of concerns [32,34]. The qualitative studies exploring the experiences of parents providing care at home [38,39,40] found that parents wanted to be supported and counseled to help them feel confident in their parenting.

#### 4.4.3. Social Support

The area of social support cannot be overlooked during transitional care. Parents who are raising premature babies at home may experience social isolation. It can become difficult for parents to leave the house, owing to the child’s immunological fragility. Their time may become limited by their caregiving duties, and they may feel socially isolated because they lack the skills to solve the various problems that arise during the process. The needs of parents experiencing transition at home include “dealing with social networks” [38], professional support for families, governmental economic support [40], and professional collaboration [39,40]. A multidisciplinary team care setting consisting of neonatal specialists, social workers, rehabilitation therapists, pediatricians, and other professionals is crucial for successful transitional care. In addition, economic support and social networks for efficient team functioning represent important aspects of transitional care in the long term.

## 5. Discussion

This study is an integrative review of the current literature regarding the topic of transitional care for parents caring for pre-term infants at home following discharge from the NICU. It was conducted to identify the main contents of transitional care for families of pre-term infants to form a basis for the development of future transitional care programs. The future directions of transitional care in South Korea are also discussed, focusing on the main attributes identified in this study.

In recent years (i.e., since 2020), the target population for pre-term infant nursing care has been expanded from just mothers to include both parents and other family members as well. Because pregnancy and childbirth are performed by women, the responsibility of childcare has also often fallen on women alone in the past. However, since the 21st century, as FCC models have been commercialized [41], the role of childcare has been expanded to include other family members as well. Since 2010, the number of studies focusing on fathers in pre-term infant research has been increasing [42,43,44], and the research subjects in such studies have been expanding to include all parents. Hemle Jerntorp et al. (2020) provided an FCC-based parent support program. This is a transitional care program that uses therapeutic communication to provide information about infant health and care. They found that fathers who experienced the transitional care program wanted to be good partners and collaborators with their spouses and healthcare professionals and felt a very strong sense of responsibility for their children [27]. Mӧrelius et al. (2021) analyzed the experiences of fathers who were involved in the breastfeeding process during FCC care. They found that fathers who participated in breastfeeding courses felt responsible for breastfeeding and wanted to provide as much support as possible [45]. This means that both the mother’s role in supporting breastfeeding and the father’s role in increasing breastfeeding rates must be considered. Therefore, when developing educational content and planning policies for transitional care, it is important to expand the scope of the audience to parents and family members and organize content that reflects their characteristics.

In most of the articles we analyzed, transitional care interventions were carried out by nurses. The main content of transitional care includes general baby care, such as feeding and bathing, as well as monitoring the health statuses of unstable patients. The expertise of trained nurses is often highly emphasized for these aspects [7,10]. However, pre-term infants have broader nursing needs than just general medical care management, rehabilitation, and development. Therefore, multidisciplinary teams—encompassing medical, rehabilitation, and social work specialists—are necessary for successful transitional care [41]. The expertise of midwives and nurses is also crucial. In South Korea, the transition period begins at the NICU discharge planning stage, but the system is still lacking. After discharge, parents need a lot of help in raising their children at home, but home visits are rarely available unless the medical condition is very serious. It is a system of periodic visits to the hospital to monitor the child’s health, which depends on the parents’ capacity. This situation causes a great deal of stress for parents and reduces their confidence as caregivers. Parents who have experienced transitional care in Korea feel that they do not receive enough medical and social support [11,12,29]. Rukeya’s (2013) systematic analysis of transitional care found that overseas, transitional care involving doctors, nurses, rehabilitation therapists, and dietitians had positive effects on attachment and increased breastfeeding rates [7]. Therefore, when developing educational content and planning policies for transitional care, it is recommended that the target audience be expanded to include fathers and other family members and that the content be structured to reflect their characteristics.

The review found that the most important domain of intervention was parenting empowerments. It comprises the following key attributes: growth and development, developmental promotion, nutrition, safe environment, general parenting and parent–infant interaction. Transitional care is developmentally supportive care, such as baby-friendly neonatal intensive care initiatives and kangaroo care, that is provided from the NICU to the home [29,35,37]. As the focus of transitional care shifts from healthcare providers to parents, it is important that parents’ parenting skills are enhanced. Parenting knowledge and skills (such as parenting knowledge and nutrition and environmental care to promote growth and development) are less culturally specific than emotional and social support. Therefore, there is an advantage in utilizing international examples to provide evidence-based practice without trial and error. As it is characterized by the voluntary participation of parents, it is necessary to develop effective educational contents by referring to various overseas cases when developing a transitional care program in Korea.

The emotional and social support domains strengthen the parental empowerment domain. Just as strong roots enable a healthy tree to thrive, adequate emotional and social support can strengthen parental empowerment (Figure 2). A key attribute of emotional support in transitional care is counselling and advocacy, which refers to support from healthcare providers to help parents manage negative emotions such as anxiety, embarrassment, and lack of confidence. The practice of emotional support in transitional care identified in this review is consistently provided through ongoing visits and phone calls for 3–12 months after discharge [18,30,36,39]. In contrast, in the Republic of Korea, emotional support provided by healthcare professionals is provided during intermittent hospital visits, and, furthermore, parents have very limited opportunities to receive support by calling the hospital. Fortunately, the rapid development of the internet is alleviating these limitations. Therefore, the internet should be utilized to increase the opportunities for interaction between healthcare providers and parents to build an emotional support system.

Social support refers to community resources that can support parents in transitional care. The main attributes are community network and professional collaboration. As this is a highly culturally and socially influenced area, it is important to understand the context of healthcare networks in the Republic of Korea. Although general community healthcare support networks are well established domestically, transitional care support systems comprising multidisciplinary professionals are still lacking. Vohr et al. (2017) used a multidisciplinary team consisting of physicians, nurse practitioners, clinical social workers, and trained family resource specialists (FRS) to provide transitional care. In particular, FRSs are not medical professionals but are trained in transitional care and provide transitional care under the supervision and guidance of a social worker. This multidisciplinary team-based transitional care resulted in a significant reduction in the rehospitalization rate of infants [32]. Based on this, specific implementation policies should be developed in Korea, such as establishing multidisciplinary transitional care delivery teams and training FRS personnel. This cannot be done by small medical institutions or individuals, so public healthcare organizations should be involved.

Finally, this study has shown that the use of smartphones in transitional care interventions is increasing. The development of the internet and the global commercialization of smartphones have had a significant impact on nursing interventions. With the rapid development of artificial intelligence programs and smartphones, it is now possible to provide sufficient care to patients without the need for face-to-face human contact. This can play a very positive role in the formation of social networks for families with premature babies. For families who are often isolated from outside society due to their child’s medical condition, a smartphone-based transitional care program can help them overcome their social isolation. Therefore, various research activities should continue to be carried out to appropriately utilize the rapidly developing artificial intelligence and smartphone technologies to improve the quality of transitional care for families with premature babies.

## 6. Conclusions and Limitation

We conducted an integrative review to summarize the current literature on transitional care for parents caring for pre-term infants at home after discharge from the NICU. Three main domains of transitional care for parents of pre-term infants at home were identified. The first was parenting empowerment, which included attributes, such as growth and development, developmental promotion, nutrition, safe environment, general parenting, and parent–infant interaction. The second was emotional support, where counseling, advocacy, and community networks were identified as key attributes. The last domain was social support, which include the attributes of professional collaboration. These results were based on South Korean nursing practice. In light of these findings, policymakers and clinical practitioners involved in the development of transitional care policies for families of pre-term infants should strive to build integrated transitional care programs that are appropriate for national contexts.

The strengths of this study include the extensive review of the literature to identify the key attributes of accurate transitional care and the integration of the findings of high-quality studies through a systematic quality assessment process, which contributes to the reader’s understanding of transitional care. It also provides a direction for the establishment of transitional care in the Republic of Korea and forms a basis for the development of programs.

However, the selection of Korean- and English-language publications limits the ability to identify the attributes of transitional care that reflect more diverse cultures. Therefore, future efforts should be made to gain insights into transitional care in Asian countries with similar cultural characteristics to the Republic of Korea.

## Figures and Tables

**Figure 1 healthcare-12-02287-f001:**
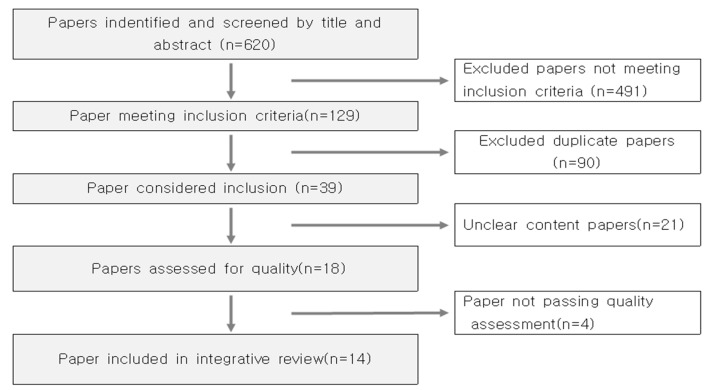
Flowchart of the search and article study selection process.

**Figure 2 healthcare-12-02287-f002:**
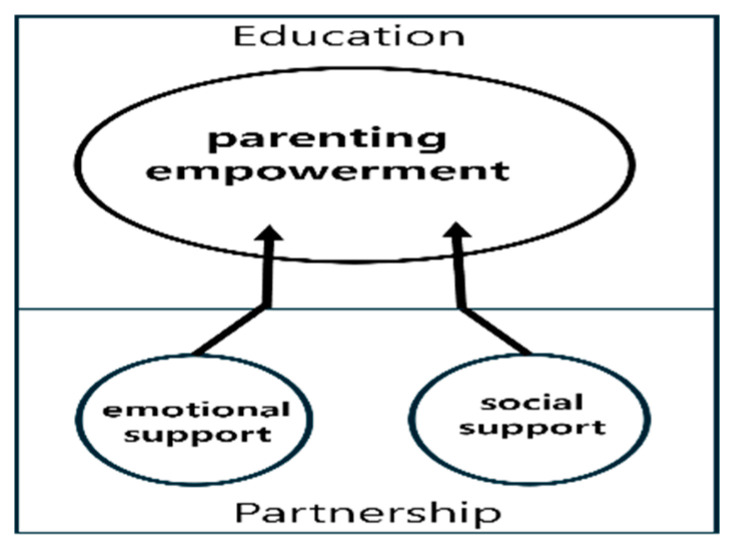
Main domain of transitional care.

**Table 1 healthcare-12-02287-t001:** The population, intervention, outcome, and timeframe (PIOT) format adopted in this study.

P	I	O	T
Families with a pre-term in their home	Key components of a transitional program needed for families at home	Health and well-being of both parents and pre-term infants	Within 3 years post-discharge from the NICU

**Table 2 healthcare-12-02287-t002:** Quality appraisal of the included articles.

Non-Randomized Studies	Participants	Measurement Outcome	Outcome Data	Design/Analysis Accuracy	Intervention as Intended
Kim et al. (2015) [29]	Y	Y	Y	Y	Y
Casey et al. (2017) [30]	Y	Y	Y	Y	Y
Poggioli et al. (2016) [31]	Y	Y	Y	Y	Y
Vohr et al. (2017) [32]	Y	Y	Y	Y	Y
**Randomized Controlled Trials**	**Randomization**	**Comparable Group**	**Outcome Data**	**Blinding**	**Adherence**
Wangruangsatid et al. (2012) [33]	Y	Y	Y	C	Y
Gund et al. (2013) [34]	Y	Y	Y	Y	Y
Landsem et al. (2015) [35]	Y	Y	Y	Y	Y
Edraki et al. (2015) [36]	Y	Y	Y	Y	Y
Robinson et al. (2016) [37]	Y	Y	Y	Y	Y
Garfield et al. (2016) [18]	Y	Y	Y	Y	Y
Phagdol et al. (2022) [19]	Y	Y	Y	Y	Y
**Qualitative Studies**	**Qualitative Approach**	**Data Collection**	**Adequacy of Results**	**Interpretation of Results**	**Coherence of Methods**
Breivold et al. (2019) [38]	Y	Y	Y	Y	Y
Hägi-Pedersen et al. (2021) [39]	Y	Y	Y	Y	Y
Haemmerli et al. (2022) [40]	Y	Y	Y	Y	Y

Y: yes; N: no; C: can’t tell.

**Table 3 healthcare-12-02287-t003:** Analysis of the included articles (n = 14).

Characteristics	Categories		n (%)
Publication year	2012–2015		5 (35.7)
	2016–2019		6 (42.9)
	2020–2022		3 (21.4)
Research design	Quantitative study	Quasi-experimental study	2 (14.2)
		Randomized controlled trial	7 (50.2)
		Cohort study	2 (14.2)
	Qualitative study	Semi-structured interview	3 (21.4)
		Content analysis	2 (14.2)
		Thematic analysis	1 (7.2)
Theoretical frameworks	Applied		2 (14.2)
	Not applied		12 (85.8)
Research subjects	Parents (mother, father)		5 (35.7)
	Mother		5 (35.7)
	Family (mother–infant, families)		4 (28.6)
Intervention provider	Nurse (e.g., neonatal, home healthcare, APN)		8 (57.2)
	Multidisciplinary team		3 (21.4)
	Other or unknown		3 (21.4)

**Table 4 healthcare-12-02287-t004:** Analyses of the selected studies.

Author (Year)	Research Design	Subjects (N)	Program			Key Findings
			Duration/ Methods	Provider	Content and Theory Frame	
Kim et al. (2015) [29]	Quasi- experimental	Mother (exper. 21/ control 22)	1 month/ home visits, telephone counseling	Home healthcare nurse	Preparing discharge infant care (feeding, bathing, touching, infant assessment, use of medical device)	Effect on maternal anxiety and confidence in role
Casey et al. (2017) [30]	Quasi- experimental	Families (exper. 234/ control 234)	12 months/ home visits	Nurse social worker	Infant care (safe sleep, crying, medication administration, formula preparation, G&D, injury prevention), social support (community services, follow-up schedule)	Effect on infant mortality rate, rates of immunization and visits to clinic
Poggioli et al. (2016) [31]	Retrospective cohort study	Parents (exper. 61/ control 62	12 months/ home visits, telephone counseling	Physiotherapist, nursing staff, neurologist, neonatologist	In NICU: NIDCAP After NICU: motor, relational, environmental, and transactional interventions	Effect on infant’s psychomotor, behavioral development
Vohr et al. (2017) [32]	Prospective cohort study	Mother (exper. 448/ control 356)	3 months/ home visits	Neonatal nurse practitioner, multidisciplinary team	Infant care (growth, feeding, and respiratory status) Mother’s comfort, concerns, and understanding of the care plan	Effect on rate of visits to clinic
Wangruangs-atid et al. (2012) [33]	Randomized controlled trial	Mother (exper. 40/ control 41)	1 month/ home visits, telephone counseling	Unknown	In NICU: NIDCAP After NICU: assessing G&D, environmental care Emergency care	Effect on parent’s perception and knowledge of infant Infant illness rate
					Theory frame: experience transition theory	
Gund at al (2013) [34]	Randomized controlled trial	Families (exper. 12/ control 32/ compare 8)	3 months/ web app, video call	Nurse	Assessing G&D, nutrition, parental coping with care in terms of communication and sharing	Effect on parental satisfaction, use of ICTs
Landsem et al. (2015) [35]	Randomized controlled trial	Mother–infant (exper. 65/ control 62/ compare 59)	3 months/ home visits	Nurse	Infant care (understanding of their child’s expressions, promoting sensitive, positive, practical transactions between parents and infants)	Effect on parents’ quality of life and anxiety, as well as infant physical health
Edraki et al. (2015) [36]	Randomized controlled trial	Parents (exper. 30/ control 30)	6 months/ home visits	Nurse	Infant care (breastfeeding, nutrition, bathing, supplementary drugs), parental counseling and support	Effect on infant’s development
Robinson et al. (2016) [37]	Randomized controlled trial	Mother–infant (exper. 47/ control 42)	3 weeks/ web app, video call	Neonatal nurses	Infant care (sleep, nutrition, spitting up, skin-to-skin care) Infant assessment (general health, activity)	Effect on parental satisfaction, rate of visits to clinics
Garfield et al. (2016) [18]	Randomized controlled trial	Parents (exper. 46/ control 44)	1 month/ smartphone app	Nurse	Infant care (tracking activities of daily living) Parental psychological support	Effect on parenting competence in parents
Phagdol et al. (2022) [19]	Randomized controlled trial	Mother (exper. 75/ control 65)	12 months/ smartphone app, YouTube	Healthcare provider	Use of smartphone apps Infant care (growth monitoring, nutrition, hygiene practice) Environmental management	Effect on knowledge of care in parents (smartphone application is an efficient alternative to community-based care)
Hägi-Pedersen et al. (2021) [38]	Qualitative study	Parents (mother 6/ father 5)	1 months/ video counseling	Neonatal nurses	Parental training and education	Comfortable early in-home care, increased confidence concerning caring for family and managing social networks
Haemmerli et al. (2022) [39]	Qualitative study	Parents (mother 20/ father 19)	6 months/ home visiting, call counseling	Advanced practice nurse	Infant care (nutrition, physiotherapy), social support, counseling, partnership	Parent’s needs: importance of continuity of care, professional collaboration
					Theory frame: FCC	
Breivold et al. (2019) [40]	Qualitative study	Mother (10)	-	-		Mother needs in home: creating a safe home environment, professional support for family, economic support from the country

FCC family centred care.

**Table 5 healthcare-12-02287-t005:** Key attributes of transitional care for families with pre-term infants.

Categories	Domain	Key Attribute
Education	Parenting empowerment	Growth and development
		Developmental promotion
		Nutrition
		Safe environment
		General parenting
		Parent–infant interaction
Partnership	Emotional support	Counseling
		Advocacy
	Social support	Community network
		Professional collaboration

## Data Availability

The data presented in this study are available from the corresponding author upon reasonable request. These data are not publicly available owing to privacy or ethical restrictions.

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
