# Peer review of "An Integrated Review of Transitional Care for Families of Pre-Term Infants"

_healthcare, 2024, doi:10.3390/healthcare12222287_

Round 1
Reviewer 1 Report
Comments and Suggestions for Authors
Dear authors, congratulations on your work. It is a current and interesting review, but it needs more detail. I list the following aspects: clarification of the inclusion and exclusion criteria—summary justification of the articles excluded after reading the full text. There is a need for greater clarification of concepts as well as a more robust discussion of the main results.
Author Response
Comments 1: clarification of the inclusion and exclusion criteria - summary justification of the articles excluded after reading the full text |
Response 1: Thank you for pointing this out. We had mentioned Inclusion and exclusion criteria were explained in lines 144 ~154, and the process of selecting the literature was explained in lines 156~162. In addition, explains the number of excluded papers such as duplicate, unclear content paper, and not passing quality assess in <Figure 1>. |
|
Comments 2: There is a need for greater clarification of concepts as well as a more robust discussion of the main results. |
Response 2: Thank you for pointing this out. We agree with this comment. We revised to emphasize this point in Disccusion area. Please check line 327~430
|
2. Response to Comments on the Quality of English Language |
Point 1: I am not qualified to assess the quality of English in this paper. |
Response 1: My thesis was written using a professional English editing service. |
Reviewer 2 Report
Comments and Suggestions for Authors
Introduction
- The introduction clearly states the background of preterm births and the need for transitional care. However, the statistical data presented (13.4 million preterm births) is a global figure, but the context of South Korea is missing. It would strengthen the argument to provide data specific to South Korea or at least the Asian region.
- Authors should incorporate regional statistics to support the relevance of transitional care in the specific geographical context.
- Example from Manuscript: The phrase "the number is steadily increasing" is vague. Quantify the rate of increase to improve clarity and provide evidence for the claim.
2. Theoretical Framework
- The authors reference the Neonatal Transitional Care (NTC) model based on Family-Centered Care (FCC) and transitional theory. While this provides a solid foundation, there is minimal critical discussion of how these theories apply specifically to South Korean practices or the cultural adaptations needed.
- A more robust critique or exploration of how the FCC and transitional theory have been adapted (or need to be adapted) for the Korean healthcare context would be valuable. For example, discuss if any specific cultural or systemic factors impact the implementation of FCC in Korea.
- The authors could elaborate on "family contact protocols" (Line 95), explaining how family involvement is culturally mediated in South Korea compared to Western contexts where FCC originated.
3. Materials and Methods
- Feedback: The integrative review methodology is well described, following Whittemore and Knafl’s model. However, the search strategy for the literature could be more explicit. For instance, the databases selected (e.g., KISS, RISS) are mostly Korean-based, with fewer international databases included.
- The authors should include more international databases such as PubMed or Cochrane to ensure a broader scope of literature, especially given the international nature of neonatal care. Moreover, explain the exclusion of key databases like PubMed.
- In Table 1 (PIOT format), the time frame (T) is “After discharge from the NICU,” which is quite vague. Defining a more specific period (e.g., within six months post-discharge) would improve the specificity of the research scope.
4. Results: Characteristics of Included Articles
- The results section provides a reasonable analysis of the included studies. However, there is a lack of critical evaluation of the quality of the studies, especially since only 14 were included. Mentioning that several studies had small sample sizes or methodological limitations but not detailing those limitations diminishes the critical depth.
- Provide a critical assessment of study limitations. For instance, acknowledge the potential bias in studies with small sample sizes or those with retrospective designs. Highlight how these limitations affect the generalizability of the findings.
- In Table 3, the “Qualitative Study” section does not explain the methods used in sufficient detail. Include more about the methodologies, such as the interview techniques or qualitative analysis approaches used in studies like those by Breivold et al. (2019) and Hägi-Pedersen et al. (2021).
5. Key Attributes of Transitional Care
- The discussion remains descriptive. There is minimal analysis of how these attributes interact with each other or how they could be prioritized based on the specific needs of preterm infants in South Korea.
- Authors should present the key attributes in a more analytical framework. For instance, suggest how emotional support and parenting empowerment interact and how one could be prioritized over the other depending on family circumstances. A visual model of the key attributes and their interrelations would greatly enhance the clarity of the manuscript.
- The section on parental empowerment (Line 253-279) lists multiple interventions without discussing their relative importance or how these interventions could be tailored for different family structures (e.g., single-parent families).
6. Discussion
- The discussion not sufficiently compare findings with international practices. South Korean transitional care is unique, and discussing how it differs from or aligns with practices in other countries would give the paper more depth.
- Compare the findings with international benchmarks in neonatal care (e.g., from OECD countries or the WHO). This would allow for a richer discussion on the cultural and systemic differences in transitional care.
- Authors can use comparative examples to illustrate the sentence "the role of child care has been expanded to include other family members" (Line 314-316) , such as how other countries have integrated fathers into neonatal care.
7. Inaccuracies
- Inconsistency: The manuscript uses different terms for "preterm infants" (e.g., "high-risk newborns" and "preterm babies"). Consistency in terminology is essential for clarity and precision.
- Inaccuracy: The study claims that transitional care leads to improved breastfeeding rates (Line 97-98). However, no evidence or reference is cited to support this claim. Include references or more data on this point.
- Misinterpretation: In the results section, the authors state that the majority of studies focused on mothers (78.6%). However, they imply that focusing on mothers is a limitation without discussing the evolving role of fathers in South Korean culture and neonatal care.
8. Conclusion
- High-impact journals like this should have conclusions that drive policy change or suggest concrete next steps.
- For example, the conclusion could also mention the need for more longitudinal studies to evaluate the long-term effects of transitional care and instead of stating, “These results will hopefully prove useful” (Line 364-365), the conclusion should include a call for action, such as “Policymakers and healthcare providers should prioritize the integration of multidisciplinary teams in transitional care programs to ensure comprehensive support for families.”
Moderate English editing needed
Author Response
Comments 1: Introduction The introduction clearly states the background of preterm births and the need for transitional care. However, the statistical data presented (13.4 million preterm births) is a global figure, but the context of South Korea is missing. It would strengthen the argument to provide data specific to South Korea or at least the Asian region. Authors should incorporate regional statistics to support the relevance of transitional care in the specific geographical context. Example from Manuscript: The phrase "the number is steadily increasing" is vague. Quantify the rate of increase to improve clarity and provide evidence for the claim.
|
Response 1: Thank you for pointing this out. We agree with this comment. Therefore, we has beed added Korea’s premature birth rate. Please check line 27~28. |
|
Comments 2: Theoretical Framework l The authors reference the Neonatal Transitional Care (NTC) model based on Family-Centered Care (FCC) and transitional theory. While this provides a solid foundation, there is minimal critical discussion of how these theories apply specifically to South Korean practices or the cultural adaptations needed. l A more robust critique or exploration of how the FCC and transitional theory have been adapted (or need to be adapted) for the Korean healthcare context would be valuable. For example, discuss if any specific cultural or systemic factors impact the implementation of FCC in Korea. l The authors could elaborate on "family contact protocols" (Line 95), explaining how family involvement is culturally mediated in South Korea compared to Western contexts where FCC originated.
|
Response 2: Thank you for pointing this out. We agree with this comment. l Therefore, we additionally described the domestic application situation of the FCC and NTC theory, and the systems and directions that need to be improved in the future. Please check line 99~109. l Additionally, since the recommended content is related to content that should beincluded in Discussions, this content was reflected in the Discussion. Please check line 334~341
|
Comments 3: Materials and Methods l The integrative review methodology is well described, following Whittemore and Knafl’s model. However, the search strategy for the literature could be more explicit. For instance, the databases selected (e.g., KISS, RISS) are mostly Korean-based, with fewer international databases included. l The authors should include more international databases such as PubMed or Cochrane to ensure a broader scope of literature, especially given the international nature of neonatal care. Moreover, explain the exclusion of key databases like PubMed. l In Table 1 (PIOT format), the time frame (T) is “After discharge from the NICU,” which is quite vague. Defining a more specific period (e.g., within six months post-discharge) would improve the specificity of the research scope.
Response 3: Thank you for pointing this out. l Initially, we used not only Korean databases but also international databases such as Google Scholar, MEDLINE, Embase, CINAHL, and Science Direct, where English-language articles can be found. We did not use PubMed in this study, but we used MEDLI, which allows you to search and view medical articles related to medicine. Please check line 135~142. l Based on your feedback, we've modified the time period (T) in Table 1 by adding 3 years or less. Please check Table 1.
|
|
Comments 4: Results
- The results section provides a reasonable analysis of the included studies. However, there is a lack of critical evaluation of the quality of the studies, especially since only 14 were included. Mentioning that several studies had small sample sizes or methodological limitations but not detailing those limitations diminishes the critical depth.
- Provide a critical assessment of study limitations. For instance, acknowledge the potential bias in studies with small sample sizes or those with retrospective designs. Highlight how these limitations affect the generalizability of the findings.
- In Table 3, the “Qualitative Study” section does not explain the methods used in sufficient detail. Include more about the methodologies, such as the interview techniques or qualitative analysis approaches used in studies like those by Breivold et al. (2019) and Hägi-Pedersen et al. (2021)
Response 4: Thank you for pointing this out. We agree with this comment.
- The quality assessment of the literature selected for this study used a validated tool called the MMAT.
- The process of selecting 14 studies for final analysis and the rationale for excluding 6 studies are further described on lines 184-188.
- Table 3 adds interview techniques and analysis methods for qualitative research. Please check Table 3.
Comments 5: Key Attributes of Transitional Care
- The discussion remains descriptive. There is minimal analysis of how these attributes interact with each other or how they could be prioritized based on the specific needs of preterm infants in South Korea.
- Authors should present the key attributes in a more analytical framework. For instance, suggest how emotional support and parenting empowerment interact and how one could be prioritized over the other depending on family circumstances. A visual model of the key attributes and their interrelations would greatly enhance the clarity of the manuscript.
- The section on parental empowerment (Line 253-279) lists multiple interventions without discussing their relative importance or how these interventions could be tailored for different family structures (e.g., single-parent families).
Response 5: Thank you for pointing this out. We agree with this comment.
- We have provided additional descriptions of the interactions between emotional and social support in parenting capacity building, as well as a visualization of the interrelationships between the key domains. Please check line 260~
- Interpretation and discussion of the meaning of the key attributes commented on and the feasibility of the interventions in different contexts were judged to be best presented in the section discussion analysing the implications of the study; they have therefore been added to the discussion section. Therefore, it was added to in Discussion. Please check line 376~428
Comments 6: Discussion
- The discussion not sufficiently compare findings with international practices. South Korean transitional care is unique, and discussing how it differs from or aligns with practices in other countries would give the paper more depth.
- Compare the findings with international benchmarks in neonatal care (e.g., from OECD countries or the WHO). This would allow for a richer discussion on the cultural and systemic differences in transitional care.
- Authors can use comparative examples to illustrate the sentence "the role of child care has been expanded to include other family members" (Line 314-316) , such as how other countries have integrated fathers into neonatal care.
Response 6: Thank you for pointing this out. We agree with this comment.
- The weaknesses and characteristics of Korea's transitional care are compared with those of other countries' transitional care, and a discussion on the future development of Korea's transitional care is added. Please check the full discussion.
- In response to your comment, we've added examples and comparisons of fathers' transitional care in other countries to make it more compelling. Please check line 342~
Comments 7: Inaccuracies
- Inconsistency: The manuscript uses different terms for "preterm infants" (e.g., "high-risk newborns" and "preterm babies"). Consistency in terminology is essential for clarity and precision.
- Inaccuracy: The study claims that transitional care leads to improved breastfeeding rates (Line 97-98). However, no evidence or reference is cited to support this claim. Include references or more data on this point.
- Misinterpretation: In the results section, the authors state that the majority of studies focused on mothers (78.6%). However, they imply that focusing on mothers is a limitation without discussing the evolving role of fathers in South Korean culture and neonatal care.
Response 7: Thank you for pointing this out. Below is a detailed response to your comment.
- Inconsistency: We've unified the terminology with “pre-term infants” throughout.
- Inaccuracy: The theoretical evidence that transitional care improves breastfeeding rates is based on the reference [18] and the reference number is provided. If you check the references, you will see that our statements are not inaccurate.
- Misinterpretation: Our findings describe that 78.6% of the 14 studies we reviewed had mothers in the study population. However, the issue you raised about “the content of fatherhood in Korean culture and the direction of change” is an important point in relation to our study, so we have added it to the Discussion section. Please check line 334~
Very grateful for your sincere advice.
Comments 8: Conclusion
- High-impact journals like this should have conclusions that drive policy change or suggest concrete next steps.
- For example, the conclusion could also mention the need for more longitudinal studies to evaluate the long-term effects of transitional care and instead of stating, “These results will hopefully prove useful” (Line 364-365), the conclusion should include a call for action, such as “Policymakers and healthcare providers should prioritize the integration of multidisciplinary teams in transitional care programs to ensure comprehensive support for families.”
Response 8: Thank you for pointing this out. We agree with this comment.
- We modified the content as per your advice. Please check line 441~444.
Reviewer 3 Report
Comments and Suggestions for Authors
This integrative review explores the key elements of transitional care for families with pre-term infants and aggregates data to support the development of relevant programs. The theoretical framework is well-constructed.
Lİne 27: The rate of preterm births should be provided for context. Without knowing the total number of births, stating a number alone is insufficient.
Lines 64-70: The aim paragraph should be rewritten more fluidly, avoiding unnecessary repetition.
While the term “nurturing environment of families” is mentioned twice, it is essential to also analyze the "key attributes of transitional care for families with pre-term infants" through the lens of UNICEF's Nurturing Care Principles.
In Table 1, both the subject and key findings are listed identically, which is problematic. For example, 13 out of 14 studies focus on education, but the subject should reflect the intervention target/area. For example, Edraki et al. (2015) examined the effects of parental counseling and support on child development, which is an important distinction. The manuscript should clarify that three studies consider the "baby" as the subject of transitional care, not the family.
Baby-friendly neonatal intensive care initiatives and kangaroo care should be included in the discussion on transitional care.
The manuscript must highlight its strengths and limitations of the study.
Author Response
Comments 1. Lİne 27: The rate of preterm births should be provided for context. Without knowing the total number of births, stating a number alone is insufficient.
|
Response 1: Thank you for pointing this out. We agree with this comment. Therefore, we Added preterm birth rate and preterm birth status in Republic of Korea. Please check line 27~28 |
|
Comments 2: Lines 64-70: The aim paragraph should be rewritten more fluidly, avoiding unnecessary repetition.
|
Response 2: Thank you for pointing this out. We agree with this comment. Therefore, we modified the content to avoid unnecessary duplication of sentences. Please check line 65~68 |
|
Comments 3: In Table 1, both the subject and key findings are listed identically, which is problematic. For example, 13 out of 14 studies focus on education, but the subject should reflect the intervention target/area. For example, Edraki et al. (2015) examined the effects of parental counseling and support on child development, which is an important distinction. The manuscript should clarify that three studies consider the "baby" as the subject of transitional care, not the family. Response 3: Thank you for pointing this out. We agree with this comment. Therefore, we modified it based on your advice. The content-related table is Table 4, and all transitional interventions are mainly focused on education and counseling. All interventions targeting are parents and families. However, in the process of organizing the literature review, we were not careful in expressing the subject. So we expressed it as an infant. Therefore, the three literatures represented by “infant” were reviewed again, the subjects of the study were family members or parents. So we modified the table 4. Please check Table 4.
Comments 4. Baby-friendly neonatal intensive care initiatives and kangaroo care should be included in the discussion on transitional care. Response 4: Thank you for pointing this out. We agree with this comment. Therefore, we added this content. Please check line 376~389.
Comments 5. The manuscript must highlight its strengths and limitations of the study. Response 5: Thank you for pointing this out. We agree with this comment. Therefore, we added the strengths and limitations of the study. Please check line 445~454
|
Round 2
Reviewer 1 Report
Comments and Suggestions for Authors
Dear Authors,
Thank you for your review of the manuscript. The issues mentioned in the previous revision are certainly more precise. However, in the “Introduction,” there is a specification to the south of Korea that is not compatible with the type of review. As this is an integrative review, the principle that should guide the justification and the problem is an approach that provides the synthesis of knowledge and the application of results from different studies. That fact concerns me, so I suggest further revision of the scope of the work.
Author Response
Comments 1: in the “Introduction,” there is a specification to the south of Korea that is not compatible with the type of review. As this is an integrative review, the principle that should guide the justification and the problem is an approach that provides the synthesis of knowledge and the application of results from different studies. That fact concerns me, so I suggest further revision of the scope of the work.
Response 1: Thank you for pointing this out. At the end of the introduction, the purpose of the study is explained, such as the need for this study. As the ultimate goal of this study is to form the basis for a study on transitional care for preterm infants in Korea, which is not yet well established in Europe, it is not unreasonable to mention the actual situation of transitional care in South Korea in the necessity section.
Reviewer 2 Report
Comments and Suggestions for Authors
what does a in bracket in line 28 mean
authors should add name of flowchart for figure 1
Table 4 with selected studies should number the studies
Line 388, what does comments 3-4 and the sign mean, if not relevant, remove
I know that PRISMA 2021 guidelines recommend documenting all eligible and excluded studies, especially for systematic and integrative reviews. Although may not be mandatory in Whittemore and Knafl’s methodology, this level of reporting can strengthen the review’s credibility and make it more comprehensive for readers and future researchers. It shows diligence in applying the quality assessment tool (MMAT) and transparency in the selection process which could improve the reproducibility of findings
Comments on the Quality of English Languagesome English language editing willn be needed before publication
Author Response
Comments 1:At the end of the introduction, the pwhat does a in bracket in line 28 mean
Response 1: Thank you for pointing this out. We agree with this comment. Therefore,the number of the added article was incorrectly entered, so wecorrectedit. Please check line28.
Comments 2:authors should add name of flowchart for figure 1
Response 2: Thank you for pointing this out. But We have already named the flowchart in Figure 1. Please check line164.
Comments 3:Table 4 with selected studies should number the studies
Response 3: Thank you for pointing this out. We agree with this comment. We insert number. Please check Table4.
Comments 4:Line 388, what does comments 3-4 and the sign mean, if not relevant, remove
Response 4: Thank you for pointing this out. There was an error during the editing process. Deleted. Thereforewe deleted it.
Comments 5:I know that PRISMA 2021 guidelines recommend documenting all eligible and excluded studies, especially for systematic and integrative reviews. Although may not be mandatory in Whittemore and Knafl’s methodology, this level of reporting can strengthen the review’s credibility and make it more comprehensive for readers and future researchers. It shows diligence in applying the quality assessment tool (MMAT) and transparency in the selection process which could improve the reproducibility of findings
Response 5: Thank you for pointing this out. We agree with this comment. Therefore
Please check line186-188 and line 573-589.
Reviewer 3 Report
Comments and Suggestions for Authors
Revisions have been made to the manuscript according to the suggestions.
However, references need to be formatted according to the journal's style guidelines. Currently, citations are given as author (year), but the journal's style does not require the year to be included.
Author Response
commet: references need to be formatted according to the journal's style guidelines. Currently, citations are given as author (year), but the journal's style does not require the year to be included.
Response: I checked the latest version of the template and found that in the references, the "year" is still written in bold after "Journal Abbreviation". This was the way it was written from the beginning, so I didn't change it.